# Evaluating the effectiveness and sustainability of a primary healthcare strategy to reduce the prevalence of strongyloidiasis in endemically infected Indigenous communities in Northern Australia

Wendy A. Page[1,2]*, David Blair[3], Karen Dempsey[1], Beverley-Ann Biggs[4,5], Jenni A. Judd[6]

1 Miwatj Health Aboriginal Corporation, Nhulunbuy, Northern Territory, Australia, 2 College of Medicine and Dentistry, Division of Tropical Health and Medicine, James Cook University, Cairns, Queensland, Australia, 3 College of Science and Engineering, James Cook University, Queensland, Australia, 4 Department of Medicine, University of Melbourne, Peter Doherty Institute for Immunity and Infection, Parkville, Victoria, Australia, 5 The Victorian Infectious Diseases Service, Royal Melbourne Hospital, Parkville, Victoria, Australia, 6 Jawun Research Centre, School of Graduate Research, Central Queensland University, Bundaberg, Queensland, Australia

* wendypage13@gmail.com

**Data availability statement:** The authors confirm that all data underlying the findings are fully available without restriction. All relevant

## Abstract

### Background

Strongyloidiasis is endemic in many remote Indigenous communities in Australia. Early diagnosis, treatment, and follow-up of chronic strongyloidiasis can prevent life-threatening clinical complications and decrease transmission in these endemic communities. The aim of this paper is to evaluate the effectiveness and sustainability of a primary healthcare strategy designed to measure and reduce the prevalence of strongyloidiasis in four remote communities in northeast Arnhem Land.

### Methodology

The primary healthcare strategy was a prospective, longitudinal, health-systems intervention designed to integrate serological testing for chronic strongyloidiasis into the Indigenous preventive adult health assessment utilising the electronic health-record systems in four Aboriginal health services. Positive cases were recalled for treatment, and opportunistic follow-up serology after six months. Results were tracked using *Strongyloides* reports generated by the electronic health-record system. This paper describes the changes in prevalence, effectiveness of treatment, and reinfection during the implementation phase, 2012–2016. An improved *Strongyloides* electronic report was developed to evaluate the effectiveness and sustainability of the intervention to the end of 2020.

data are within the paper and its Supporting Information files.

**Funding:** The author(s) received no specific funding for this work.

**Competing interests:** The authors have declared that no competing interests exist.

## Principal Findings

During the entire period 2012–2020, 84% (2390/2843) of the resident adults in the four communities were tested for strongyloidiasis at least once. Prevalence was reduced from 44% (1056/2390) ever-positive to 10% (232/2390) positive on their last test. Of positive, treated cases with a follow-up serology test, the last test was negative in 85% (824/967) of individuals. Point prevalence continued to decrease in each community four years after the end of the implementation phase.

## Conclusions

The results provided practice-based evidence of a significant decrease in the prevalence of strongyloidiasis attributable to the strategy which could be replicated in other health services utilising electronic health-record systems. The final evaluation demonstrated the sustainability and ongoing benefits for endemically infected communities, and the key role that health services can play in strongyloidiasis prevention and control programs.

## Author summary

This primary healthcare strategy originated in an Aboriginal community-controlled health organisation in remote Northern Territory communities in Australia where strongyloidiasis is endemic. The goal was to prevent clinical complications and reduce prevalence of strongyloidiasis. Serological testing for chronic strongyloidiasis was integrated into the preventive adult health assessments in four Aboriginal health services. Positive cases were recalled for treatment, then follow-up serology after six months.

Data extracted from the electronic health record (EHR) system during the implementation phase, 2012–2016, measured changes in prevalence, treatment effectiveness, and reinfection. An evaluation in 2020 used an improved *Strongyloides* report to extract data from 2012–2020. This showed that 84% of the adult population in the four communities were tested for strongyloidiasis, with prevalence decreasing from 44% to 10%. Of treated cases with follow-up tests, 85% had negative results.

These results showed a significant reduction in the prevalence of strongyloidiasis, highlighting the effectiveness and sustainability of the intervention. The *Strongyloides* report provided a monitoring and ongoing surveillance tool that was accessible to clinicians, as well as de-identified reports for the local health board and communities. This research highlights the important role of health services in prevention and control programs for strongyloidiasis in endemic communities.

## Introduction

Strongyloidiasis is a preventable and treatable chronic infectious disease caused by *Strongyloides stercoralis*, an unusual microscopic nematode with a unique life cycle involving autoinfection and long-term persistence in the human host, with the risk of potentially fatal complications [1,2]. An estimated 614 million people are infected with *S. stercoralis* worldwide [3]. In 2020, strongyloidiasis was added to the World Health Organization's Neglected Tropical Diseases (NTDs) Roadmap 2021–2030, which sets goals to prevent, control, eliminate, or eradicate NTDs [4]. One of the targets of this roadmap is to develop effective control programs for strongyloidiasis [5]. To be effective and sustainable, control programs need to measure prevalence in the target population, implement, monitor and evaluate an intervention, and provide ongoing surveillance to prevent resurgence. These interventions need to be tested in real-world settings and the results documented [5,6].

Australia is considered a non-endemic country for strongyloidiasis, but endemic hotspots exist in remote Indigenous communities, where prevalence has been as high as 60% [7–10], and local reinfection can occur [11]. Following the first National Workshop on strongyloidiasis in 2001, Aboriginal leaders requested action, stating they did not want to suffer from a preventable disease that is rare in mainstream Australia [12,13]. Subsequent studies have emphasised the need for community-driven and ongoing control programs [14,15]. However, a population health-systems approach or policy for the control and prevention of strongyloidiasis in endemic communities in Australia is still lacking [16].

Although *S. stercoralis* is categorised as a soil-transmitted helminth (STH), its unique autoinfective life cycle has considerable implications for control programs [17]. For example, *S. stercoralis* does not rely solely on soil for transmission [17,18], and persons with disseminated strongyloidiasis are highly infectious through faeces and sputum [19,20]. Faecal examination detects *S. stercoralis* larvae rather than eggs, and are essential tests for unwell, immunocompromised patients for whom serology may be negative [2] but are less likely to detect larvae during the chronic phase [21]. In contrast, serological testing has high sensitivity and specificity in the chronic phase [22,23]. The goal of treatment for *S. stercoralis* is eradication, as compared to the reduction of worm load for other STHs. Follow-up serology is useful in measuring the effectiveness of treatment [24–26] and changes in prevalence [27].

In Australia, *S. stercoralis* serology was integrated into the refugee health assessment in 2010 to prevent clinical complications and transmission in a non-endemic country [28,29]. Despite this policy for refugees and acceptance of serology and treatment for individuals returning to non-endemic locations, there remains some uncertainty about the usefulness of serological testing in endemically infected communities where reinfection can occur [8,30]. Four previous studies in Australian Indigenous communities have shown that the Strongyloides-specific IgG antibodies decline following effective treatment, and that serology is useful for measuring changes in prevalence in endemic hotspots [11,15,24,31]. However, these studies had limited timeframes. High turnover of clinicians in remote communities and low awareness of this most neglected tropical disease also contributed to the lack of ongoing surveillance [32–35]. Thus, there is a need for a sustainable systems-based approach that embeds testing, treatment and follow-up into existing clinical practices, and tracks prevalence over time [14,36].

The Australian government has a national strategy for closing the gap in health inequity between Indigenous and non-Indigenous Australians [37]. Primary healthcare services play a crucial role, particularly in remote Indigenous communities in encouraging annual preventive health assessments, which aim for early diagnosis, treatment, and management of common conditions that contribute to morbidity and early mortality in Indigenous communities [38,39]. These assessments are conducted by local health services, and the data is entered into the electronic health record (EHR), which also enables accessible detailed population health reports [40].

The primary healthcare strategy designed for this research was initiated in July 2012, when *S. stercoralis* serology was added to the adult preventive health assessments in four Aboriginal health services. This intervention integrated the diagnosis, treatment, and follow-up of positive cases of chronic strongyloidiasis within a preventable chronic disease

framework. Page, Judd, MacLaren, & Buettner (2020) reported the significant increase in coverage and detection of positive cases of strongyloidiasis during the four-year implementation phase [41]. The current paper aims to determine if there was evidence of a decrease in the prevalence of strongyloidiasis attributable to the intervention during 2012–2016, and to evaluate the 2012–2020 data for evidence of sustainability of the program beyond the implementation period.

## Methods

### Ethics approval

As this paper describes the second part of the population health-systems intervention in Page et al, 2020 [41], the ethics approval is the same:

Ethics approval was gained from the Human Research Ethics Committee of the Northern Territory Department of Health and Menzies School of Health Research (HREC Reference Number 2012-1868) and James Cook University (HREC Approval number H4953). The Board of Miwatj Health Aboriginal Corporation, the owners of the clinical databases with responsibility for deciding what research is appropriate and acceptable, provided written approval for this study. Individual adult patients provided oral consent for a preventive health assessment that was documented in their electronic health record. All data extracted for this study was de-identified.

### Study design

This *Strongyloides* control strategy was designed as a prospective, longitudinal, health-systems intervention, aligning with existing population-health strategies and reporting systems within four Aboriginal health services in endemically infected communities. This research study commenced implementation in 2012. This paper focuses on measuring the changes in prevalence: (i) during the implementation phase at half-yearly intervals for each health service until December 2016, (ii) a comprehensive analysis of data collected from 2012–2016, including those with follow-up serology test results, and (iii) a final evaluation in 2020 using an improved *Strongyloides* report assessed the effectiveness and sustainability of the intervention in these endemically infected communities.

### Key parameters

Three key parameters were measured for this study. Coverage was the number of persons as a proportion of the resident adult population who were tested at least once for strongyloidiasis. Period prevalence was the number of persons testing positive as a proportion of those tested, also referred to as ever-positive. Point prevalence was the number of persons testing positive on their most recent test as a proportion of those tested, at the time of data extraction. (Fig 1).

### Participants and setting

This primary healthcare strategy was implemented in the same four remote Aboriginal health services as described by Page et al, 2020 [41]. The participating health services varied in location and size and were part of the regional Aboriginal community-controlled health organisation in northeast Arnhem Land, Northern Territory, on Yolngu country. The area served by each health service had a core resident community, although the population was dynamic, with some fluctuations through time. The four participating health services are referred to as Clinics A, B, C, and D.

In these remote communities, the roles of health service staff are broad, ranging from managing and evacuating acutely unwell patients, to outreach activities for prevention and earlier detection and management of treatable conditions that contribute to morbidity and early mortality. The high turnover of health practitioners makes the EHR vital for the continuity of patient care and population health strategies.

Health education strategies and resources were developed and used prior to the study as part of the engagement and co-design process, and also throughout the study to educate both clinicians and patients.

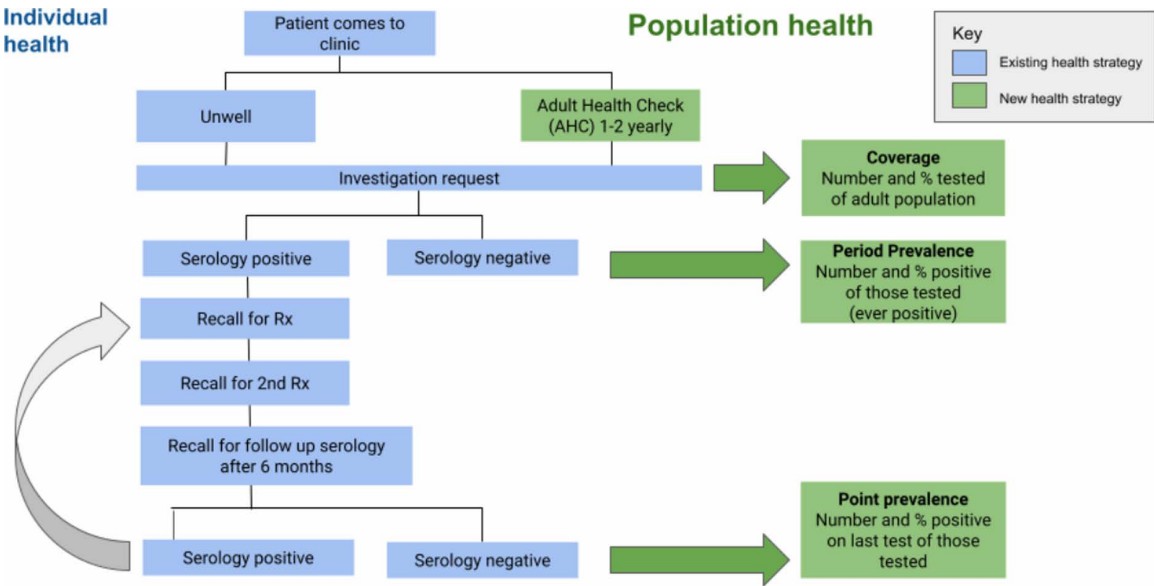

**Fig 1. Concept design for the health-systems intervention for chronic strongyloidiasis.**

## Data collection

**Selection criteria.** The selection criteria for this study were the same as those used for the preventive health assessment, also known locally as the adult health check (AHC). Indigenous adults aged 15 years and older who were current residents of the community at the time of data collection were included. A sample-size calculation prior to the study showed that 94 people per clinic were required to be tested for statistical significance. However, in this real-world setting, the first aim of the intervention was to increase coverage and identify as many infected persons as possible. The actual sample varied from 172 persons tested in Clinic A to 738 persons tested in Clinic D, and coverage increased to a combined total of 1686 persons tested by 2016, as described in Page et al [41]. Coverage to the end of 2020 is reported in this second paper.

**Diagnostic test.** Data collection for all reports commenced from 01 July 2012. This aligned with the introduction by Western Diagnostic Pathology (Perth, Western Australia) of the IVD enzyme-linked immunosorbent assay test (ELISA) (DRG Instruments GmbH, Marburg, Germany) which was used for all testing in this study [41]. This test detects anti-*Strongyloides* IgG antibodies using *S. stercoralis* L3 filariform larval antigens to measure the immune response to the autoinfective filariform larvae. An optical density (OD) of 0.2 or greater was regarded as positive, indicating chronic strongyloidiasis. An OD less than 0.2 was reported as negative and, in the context of a follow-up serology, was regarded as no longer infected. The IVD ELISA test has a sensitivity of 92.3% and specificity of 97.4% [22].

## The electronic health-record system

The flow diagram below highlights the steps for implementing the intervention in the EHR, as used in the participating health services (Fig 2). This concept could be adapted to other EHR systems as needed. Further details are provided in the previous paper by Page et al [41].

## *Strongyloides* reports

The EHR facilitates the creation of reports which allow each health service to access their own identifiable data for clinical purposes. For this research, de-identified *Strongyloides* reports were generated half-yearly during the implementation

**Fig 2. Implementation of the intervention within the electronic health-record system.**

phase from 2012 to 2016. These tracked the changes in point coverage and point prevalence of persons tested, based on the current residents at the time of data extraction [41]. The de-identified data for all persons tested from 2012 to 2016 were then exported through MS Excel into Stata 11 for further analysis, which included measuring changes in prevalence, effectiveness of treatment, and reinfection. As part of the analysis feedback and continuous quality improvement processes in these health services, the *Strongyloides* report was then improved to evaluate the strategy by facilitating efficient comparison between period prevalence and point prevalence, as well as measuring changes in point prevalence.

**Outcome measures**

The changes in prevalence in each of the four participating clinics were measured in three stages. First, the changes in point prevalence were obtained directly from the half-yearly reports for each clinic and presented with exact binomial 95% confidence intervals (CIs) from December 2012 to December 2016. Each reporting period commenced in July 2012 and included only current resident Indigenous persons aged 15 years and over at the time of data extraction. This paper describes these changes in point prevalence, whereas changes in point coverage were described in the previous paper [41].

Second, a more comprehensive analysis was conducted using Stata release 12.1 (Stata 2011) enabling period prevalence spanning 2012–2016 to be compared with point prevalence at the end of 2016. All persons who had been tested at

least once were included, stratified by clinic and overall. For each clinic and overall, period prevalence and the prevalence according to the last record were calculated and presented with exact binomial 95% CIs. Those who were tested more than once were assessed for effectiveness of treatment and for possible reinfection. Overall rates were cluster-adjusted assuming the clinics were primary sampling units. For each clinic and overall, period prevalence of persons testing positive at least once and prevalence of strongyloidiasis according to the last record were compared for differences in sex using exact Fisher's tests and for differences in age using non-parametric Mann-Whitney tests.

The third stage of this research was an evaluation to ascertain if the population health-systems intervention had been sustained beyond the initial implementation phase, and to further assess prevalence changes over the entire 8.5 years since commencement. This stage utilised the improved report which contained a *Strongyloides* prevalence indicator (SPI) to enable immediate comparison between period prevalence (ever-positive from 01 July 2012 to 31 December 2020) and point prevalence for current residents on 31 December, 2020. The numbers and percentages for coverage, ever-positive, and positive on last test were calculated and presented with exact binomial 95% CIs. During this stage, the number and proportion of seropositive persons with more than one test were analysed to assess the effectiveness of treatment. In addition, the point prevalence at the end of this evaluation phase in 2020 was compared with the point prevalence for current residents at the end of the implementation phase in 2016.

## Results

The results of this study were divided into three parts. Firstly, the point-prevalence data for each of the four clinics were tracked during the implementation phase (01 July 2012–31 December 2016). Next, a detailed analysis of extracted data for all tested persons included a comparison of period prevalence and point prevalence for the implementation phase. Finally, the evaluation measured the changes in prevalence over the whole study (01 July 2012–31 December 2020) to assess the sustainability and effectiveness of the intervention.

### Half-yearly reports extracted during the implementation phase 2012–2016

Data were collected at half-yearly intervals during this phase using the *Strongyloides* report on the electronic health-record system (S1 Table). The current resident population at each community varied at each half-yearly report for this real-world dynamic population. Changes in point prevalence occurred across each clinic as previously positive persons had follow-up serology, and new persons were tested. These changes were tracked over time for each clinic and are displayed as a trend graph with 95% CIs. (Fig 3 and S1 Table). The figure shows the point prevalence of strongyloidiasis in four Aboriginal health services as the percentage of current Indigenous residents aged 15 years and over who tested positive for strongyloidiasis, based on their last recorded result. The denominator was the number of persons tested from July 2012 to the date the report was extracted (S1 Table). The vertical lines represent the 95% confidence intervals. Clinic D joined the intervention in January 2015, and prior to this date, usual practice there was based on only testing and following-up individuals if there was a specific clinical indication.

### Analysis of data from the implementation phase 2012–2016

At the end of the implementation phase (December 2016), a comprehensive analysis of the data collected from all 1686 persons who had been tested at least once enabled a comparison between period prevalence and point prevalence. Six hundred and eighty persons (40%) tested positive for strongyloidiasis at least once during this period. By the end of 2016, 360 (21%) of these persons remained positive based on their most recent test results. Changes in prevalence in each clinic and overall provided evidence of the effect of the intervention at the end of 2016. Prevalence declined in each clinic. Clinic A declined from 51% ever-positive to 19% at last test; Clinic B declined from 38% to 15%; Clinic C declined from 48% to 26%, and Clinic D declined from 34% to 23% (Table 1). This statistical analysis included all persons tested at least

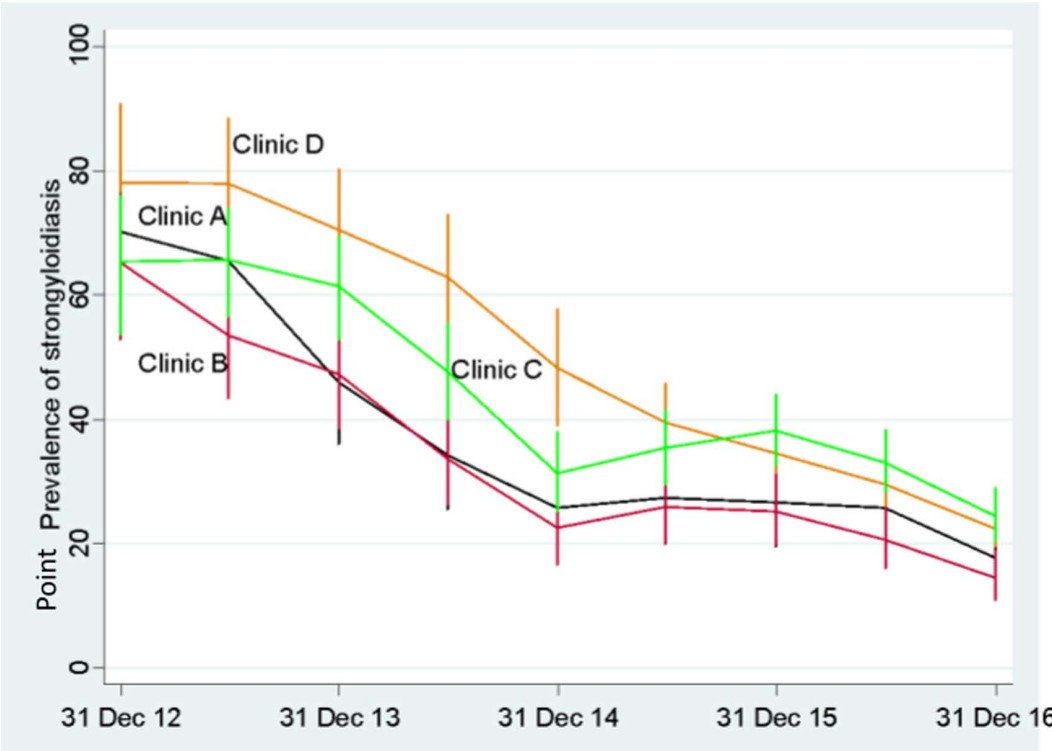

**Fig 3. Changes in half-yearly point prevalence of strongyloidiasis by health service (2012–2016) for current residents at the time of data extraction.**

once, including 77 persons who were no longer residents in communities at the end of 2016. More detail including coverage is shown in previous paper by Page et al [41].

### Effectiveness of treatment

To measure effectiveness of treatment, a further serological test was recommended 6 to 12 months after treatment. During the implementation phase, 253 of the 680 persons who tested positive were only tested once and could not be assessed for effectiveness of treatment. Thus, 427 persons were assessed for effectiveness of treatment and of these, 320 became seronegative following treatment. Treatment was therefore demonstrably effective for 75% (320/427) of persons who had a further test (S2 Table)

### Reinfection and new infections

Of the 427 positive cases with more than one test, 33 had positive-to-negative-to-positive serology results, which suggests effective treatment, followed by reinfection. Of the 33, 16 re-infected persons received further treatment and testing, and were negative on their last recorded test. The remaining 17 reinfected persons were still positive on their last result (S3 Table). In addition, 13 persons who were initially negative became positive on a later test, showing a new infection. The ongoing protocol is for any positive result, the person is recalled for treatment. These results highlight the need for an ongoing program that identifies and treats both recurring and new infections.

PLOS Neglected Tropical Diseases

**Table 1. Results at the end of 2016 for four clinics and combined totals for all persons tested at least once, period prevalence and point prevalence.**

|  | Clinic A | Clinic B | Clinic C | Clinic D | Total |
|---|---|---|---|---|---|
| Number tested at least once between 2012 and 2016; | 172 | 330 | 446 | 738 | 1686 |
| N (%) positive for *Strongyloides* at least once between 2012 and 2016; 95% CI^ | 88 (51.2%); 43.4 to 58.9 | 126 (38.2%); 32.9 to 43.7 | 214 (48.0%); 43.3 to 52.7 | 252 (34.1%); 30.7 to 37.7 | 680 (40.3%); 27.1 to 53.5 |
| N (%) positive for *Strongyloides* on last recorded test in 2016; 95% CI^ | 32 (18.6%); 13.1 to 25.2 | 48 (14.5%); 10.9 to 18.8 | 114 (25.6%); 21.6 to 30.0 | 166 (22.5%); 19.5 to 25.7 | 360 (21.4%); 14.6 to 28.1 |

^95%CI = 95% exact binomial confidence intervals.

## Demographics – gender and age

The period prevalence and the prevalence according to the last record were not statistically significantly different between the sexes. No sex difference was found between persons who remained positive or changed to negative (p = 0.878; Fisher's exact test) on last record, or between persons with or without reinfection of strongyloidiasis (p = 0.722; Fisher's exact test).

Persons who stayed positive were significantly younger (median age 31 years; IQR = [21,42]) compared with persons who became negative (median age 39 years; IQR = [28,51]; p < 0.001; Mann-Whitney test). Reinfected persons were on average older (median age 42 years; IQR = [31,50]) compared with persons with no reinfection (median age 35 years; IQR = [24,47]; p = 0.032; Mann-Whitney test.

## Final evaluation 2012–2020

A final evaluation was conducted in 2020, four years after the end of the implementation phase using an improved *Strongyloides* report to extract data on the current resident adult population at each health service. Coverage, period prevalence and point prevalence were calculated from 01 July 2012 until 31 December 2020 (S4 Table). Coverage increased to 84% (2390/2843) over this period. Changes in prevalence were measured by the number and proportion of tested persons who were positive at least once during the 8.5-year timeframe (period prevalence) compared with the number and proportion of tested persons who were positive based on their last result (point prevalence) (S4 Table).

In December 2020, the overall period prevalence was 44% (1056/2390), whereas the point prevalence had declined to 10% (232/2390). Clinic A period prevalence declined from 56% (117/209) to 4% (9/209) point prevalence, Clinic B declined from 43% (132/310) to 6% (19/310), Clinic C declined from 53% (330/625) to 13% (80/625), and Clinic D declined from 38% (477/1246) to 10% (124/1246) (Fig 4 and S4 Table). These declines were much improved compared with the analysis at the end of the 2012–2016 implementation phase of the study, when the overall period prevalence was 40%, and the point prevalence had declined to 21% (Fig 4, and Tables 1 and S4).

## Changes in point prevalence from 2016 to 2020

Point prevalence continued to decline during the four years after the end of the implementation phase (2012–2016) for each health service (Fig 4). In addition, Clinic A point prevalence declined from 18% (29/164) to 4% (9/209); Clinic B declined from 15% (45/311) to 6% (19/310); Clinic C declined from 24% (100/410) to 13% (80/625); and Clinic D declined from 22% (162/724) to 10% (124/1246) (S1 and S4 Tables). Thus, comparing changes in point prevalence from 2016 to 2020 for current residents at the time of data collection was also useful in assessing if the prompt for strongyloidiasis testing remained within the health system and if there were ongoing benefits.

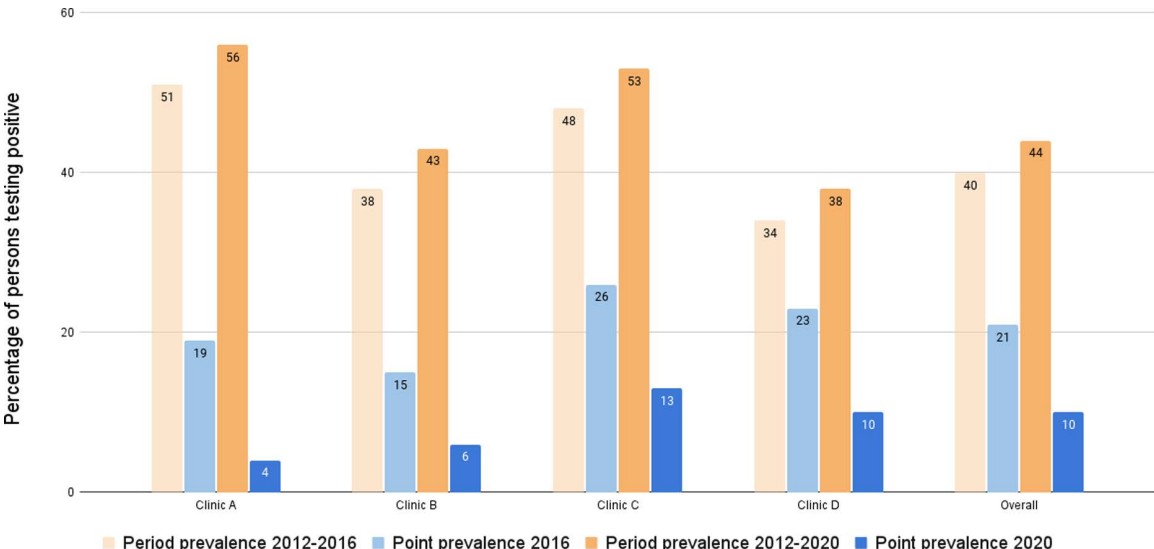

**Fig 4. Change in prevalence — comparing period prevalence to point prevalence in 2016 and 2020.**

### Effectiveness of treatment, 2020

The following flowchart shows the outcomes for the 1056 persons who tested positive at least once. The number and proportion of positive cases with more than one test also increased from 427 of 680 (63%) in 2016 to 967 of 1056 (93%) in 2020. Treatment was demonstrably effective for 824 of the 967 persons with more than one result (85%), and a further increase from 75% in 2016 (Fig 5 and S2 Table). This also provides further evidence that *Strongyloides*-specific IgG antibodies decline with effective treatment, and that follow-up serology for assessing both treatment and intervention outcomes is crucial. Comparative figures from the implementation phase (in parentheses) demonstrate improvements as a result of the ongoing program since 2016.

### Discussion

This research provided practice-based evidence that the health-systems intervention in four community health services resulted in a significant and measurable reduction in the prevalence of strongyloidiasis. The study demonstrated that *S. stercoralis* serology is a valuable tool for measuring changes in prevalence, as positive cases declined to negative with effective treatment in these endemic communities. Of those who had follow-up serology, 75% had declined to negative by 2016, and 85% by 2020. Reinfected cases also benefited from the ongoing program with re-treatment.

For the final evaluation from 2012 to 2020, the *Strongyloides* report within the electronic health- record system was improved to include current residents who had ever tested positive (period prevalence). This report demonstrated that coverage had increased to 84% of adult residents tested at least once, and period prevalence reduced from 44% ever-positive, to point prevalence of 10% positive on their last test. This report enabled an efficient assessment of the intervention's sustainability and effectiveness that showed the intervention remained embedded in the existing preventive practices four years after the end of the implementation phase. *Strongyloides* prevalence continued to decline in the four communities, demonstrating the ongoing benefits and sustainability of the program. This improved report became known as the *Strongyloides* prevalence indicator, or SPI, an accessible tool for monitoring, evaluating, and ongoing surveillance.

The ongoing approach used in this intervention meant that reinfections, recrudescence of infections, and new infections were treated and followed up in the primary health-care setting. Reinfection is a reality in endemically infected

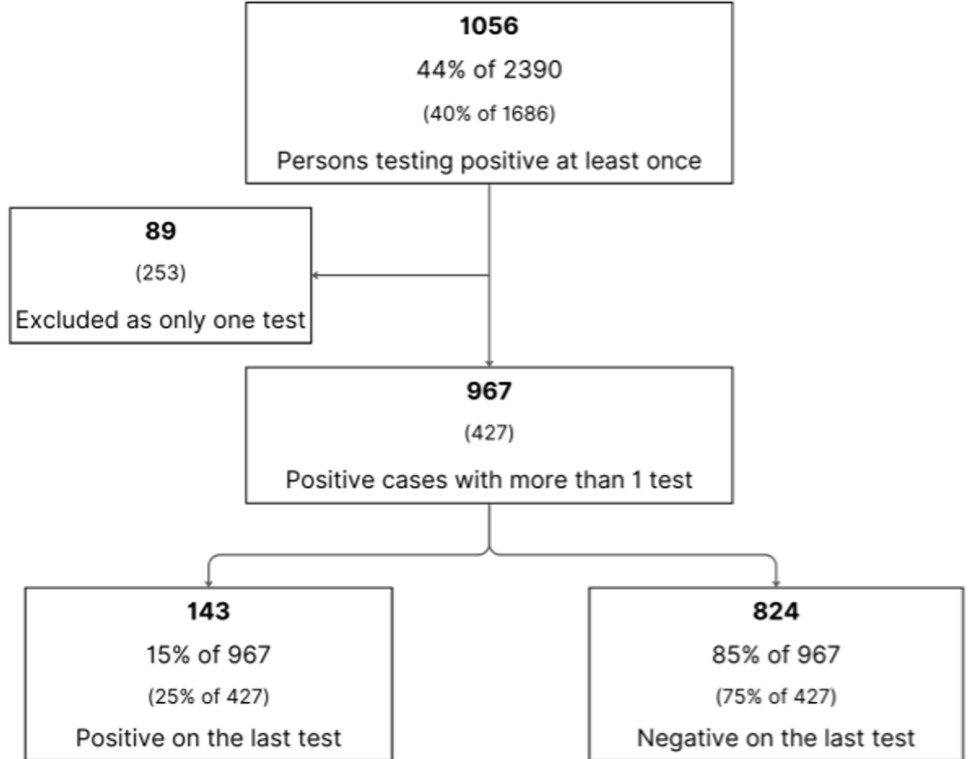

**Fig 5. Flow chart of outcomes for 1056 seropositive persons from 2012 to 2020.**

communities, and recrudescence is also a possibility if treatment has not eradicated every parasite, as even a single surviving parasite can re-establish the unique auto-infective life cycle [36,42].

### Comparison to other studies

Serological testing for strongyloidiasis is acknowledged as a useful tool to monitor the decrease in antibody levels after effective treatment in non-endemic settings [25,26,29,43]. However, evidence is limited for endemic areas, both in Australia and elsewhere [11,15,24,27,31]. This study provides further practice-based evidence of the usefulness of serology for measuring changes in prevalence in endemically infected communities. Unlike earlier studies of limited duration, this research integrated serological testing into the existing routine preventive health assessment program and electronic health-record system. This novel strategy created the SPI to efficiently measure changes in *Strongyloides* seroprevalence.

Of the four previous Australian studies in Indigenous communities, two used a mass drug administration (MDA) approach [11,15], while the other two used selective chemotherapy [24,31], treating only cases with positive tests. While both approaches can be effective, MDAs can be considered vertical control programs, as external support is often required, and is over a limited time frame [44]. Following one of these MDA studies, Kearns et al (2017) recommended an annual MDA for strongyloidiasis and scabies [11], but this was not funded. In contrast, the health-systems intervention presented here embedded an ongoing control program into the existing preventive health practice, using selective chemotherapy. This horizontal approach added strongyloidiasis to the serological testing for other conditions that cause morbidity and early mortality, such as diabetes, chronic kidney disease, hypercholesterolaemia and sexually transmissible diseases. In Australia, *Strongyloides* serology is also less costly than ivermectin, the standard drug used for treatment. Selective chemotherapy was therefore practical, efficient, and cost-effective.

## Generalisability

This paper shared the chronological research journey, which may be useful to others planning a similar program. The health-systems intervention was designed for a real-world, primary healthcare setting. The methods are shared here for others to replicate or adapt to their own setting, although results would vary in different communities with different risks for reinfection. Each clinic in our study was compared to itself, with contemporary data. The results from each of the four health services provided evidence of the scalability of this health-systems intervention, using either the original report or the improved report. Even without the comparative period prevalence data provided by the SPI in the improved report, the changes in point prevalence clearly demonstrated the benefits of an ongoing program in each clinic.

Although this intervention was designed for a known endemic community, the *Strongyloides* report can also be used to determine if an intervention is needed. For example, Clinic D did not integrate testing into the AHC until January 2015. Their first data report in 2012, when clinicians were only testing individuals if there was a clinical indication, showed the highest point prevalence of 78% (25 of 32 tested). Although that represented only 2.4% (32/1309) of their adult resident population (S1 Table), and initially was not sufficient for the sample size required in this study, it provided evidence that the population health-systems intervention was warranted in that community.

## Clinical relevance

A key role of community health services is primary prevention which involves early detection of chronic and infectious diseases, preferably when persons are asymptomatic and before clinical complications occur or further transmission occur. In comparison, the clinical protocol for secondary prevention aims to prevent iatrogenic hyperinfection and deaths, and focuses on identifying unwell, infected patients who are to be immunosuppressed [2]. However, this is dependent on clinician awareness, and many cases remain undiagnosed [18,35,45]. Embedding *Strongyloides* testing and recall reminders in primary care clinic EHR is an efficient tool for ensuring effective population coverage in endemic areas, and for overcoming the challenges of high turnover of clinicians and clinician cognitive overload [45].

The local health services provide the continuity of care that this intervention relies on. The anti-*Strongyloides* antibody tests take time to decline following treatment. Although other studies have used a decline in serology to 0.6 of initial titre at six months as evidence of cure [26,46], effective treatment in this study was assumed to have occurred only if follow-up titre declined to below the optical density cut-off value (below 0.2) reported as negative in this study. For pragmatic reasons, in this study, a positive case is treated or retreated, if the result has not declined to negative.

In this real-world health service, the priority was treatment. However, the recall for follow-up serological testing after six months was opportunistic and variable and could be delayed until the next visit to the clinic or the next AHC, particularly for young people and others who are less likely to attend the clinic regularly. For example, at the end of the implementation phase in 2016, 253 persons testing positive had only had one test, noting that the largest clinic (D) had only been implementing the intervention for two years. By 2020, only 89 persons overall had not yet had a follow-up test, showing the importance of an ongoing and longitudinal program and the value of the continuity of care from a local health service.

## Public health implications

The prevalence data has important implications for a public health response and for identifying remote communities with a high burden of disease and the need to also focus on primordial prevention to eliminate strongyloidiasis. The presence of ongoing strongyloidiasis or reinfection is an indication of the need to address environmental health issues, notably access to clean water and sanitation [47] or other sources of infection. Elimination of diseases linked to poor health infrastructure requires intersectoral collaboration between health services, government, and animal and environmental health agencies, such as a One Health approach [17]. Making strongyloidiasis nationally notifiable would also assist in identifying the hotspots that need targeted interventions [47,48].

## Strengths

This novel primary healthcare strategy meets the criteria for an effective control program as it measures prevalence, targets the adult population with chronic infection, implements, monitors and evaluates the intervention, and importantly provides ongoing surveillance. The research design is pragmatic, building on existing primary healthcare strategies in a real-world setting, and integrating into clinical practice. The EHR system enabled the efficient collection of valid and reliable data to create an effective and ongoing measurement tool to track changes in coverage and prevalence, and then collated it into the report that included the novel *Strongyloides* prevalence indicator (SPI). The SPI provided objective data on changes in prevalence as a result of the intervention, confirming that serological tests combined with selective chemotherapy are effective control strategies in these endemic communities.

The use of serology as a diagnostic tool is another major strength of this research. The results demonstrated that *S. stercoralis* serology does distinguish between current chronic infection and past (treated) infection in endemic communities, which challenges previous preconceptions [8,30]. The anti-*Strongyloides* IgG antibodies decline with effective treatment over several months, so this serological test is suited to screening, diagnosis, and follow-up. Serology is also more culturally acceptable, practical and sustainable than collecting multiple faecal specimens.

Another major strength of this research is the integrated and ongoing nature of this primary healthcare strategy, supported by the available digital health tools in a real-world setting. This distinguishes it from other past studies, which were short term, and often utilised external staff and structures. Ongoing surveillance using accessible contemporary prevalence data embedded into clinic workflow makes the program sustainable and cost effective, preventing severe clinical complications resulting in costly evacuations and hospital management [20,49].

In addition, as the human reservoir of infection decreases, the ongoing system allows greater focus on primordial prevention, such as environmental factors, which contribute not only to strongyloidiasis, but also other diseases such as acute rheumatic fever that are prevalent in these areas of disadvantage. This method is also replicable and scalable in other endemically infected populations with similar health infrastructure. The concept could also be adapted to other health settings that require measurable prevalence data relevant to the local community or high-risk group.

## Limitations

This strategy integrated *Strongyloides* serological testing into the existing Indigenous Adult Health Check, and therefore the data only includes resident Indigenous adults. Children were not included as venepuncture is not part of the child health assessment, and point-of-care tests using dried blood spots are not yet available in these settings [50]. Under the current deworming program, children aged 5–15 years receive albendazole, which helps in reducing worm load, but eradication of all *S. stercoralis* phases is required to prevent recrudescence from the autoinfective life cycle.

Although this study did not collect data on treatment, the change in serological results from positive to negative indicated that treatment with Ivermectin was 85% effective. This outcome aligns with the findings of other studies, where Ivermectin has been shown to be 86% effective [51,52].

This population health-systems intervention did not have the capacity to investigate de-identified individuals. The smaller number of remaining positive individuals could be further researched for reasons such as adherence to treatment, inadequate treatment, resistance to treatment or investigating sources of environmental reinfection. Those individuals not responding to multiple doses of ivermectin may be suited to second-line future treatment options such as ivermectin/albendazole combination or moxidectin.

Another limitation was the capacity of health staff to conduct these preventive health assessments while also managing acute and unwell patients in areas with a high burden of disease. The high turnover and shortages of health professionals in remote areas is an ongoing challenge [32,53], and there was no funding for additional staff for this program. Despite these limitations, this program has substantially decreased the prevalence, but not eliminated strongyloidiasis.

### Future recommendations

The improved *Strongyloides* report developed for this research provides a valuable source of data for further research. It allows local clinicians access to quantitative, contemporary and identifiable data and serves as an accessible register for clinical research, comparable to other existing disease registers. De-identified prevalence data can be used to identify hotspots that require targeted interventions, or to inform public health programs. Australia is known to have hotspots but lacks a system to identify them. Making strongyloidiasis nationally notifiable would help identify areas of need [47,48].

This novel strategy was very effective in measuring and reducing the prevalence of strongyloidiasis in the target population. However, primary prevention alone is not enough to eliminate the parasite in these endemic communities. Elimination requires intersectoral collaboration to address primordial prevention, such as a One Health approach that considers human, animal, and environmental sources of transmission [17].

## Conclusions

The primary healthcare strategy developed for this research resulted in a significant reduction in prevalence of strongyloidiasis between 2012 and 2016. This decline is attributed to the integration of *Strongyloides* testing into routine Indigenous health checks and subsequent management of positive cases. The continuing decrease in seroprevalence through to 2020 provides evidence of the intervention's ongoing effectiveness and sustainability within the context of remote Indigenous health services.

This study demonstrates the value of an integrated primary healthcare approach to control of chronic strongyloidiasis, framing the infection as a preventable and treatable chronic infectious disease. The results strongly support the feasibility and efficacy of serological testing, and the benefits of long-term control programs in endemic populations. The research offers practice-based evidence to inform the development of coordinated and sustainable *Strongyloides* control programs for endemically infected populations both in Australia and beyond. While *Strongyloides stercoralis* poses significant challenges, particularly within disadvantaged communities, this study highlights the critical role of such interventions in addressing health inequities and advancing progress towards closing the gap on strongyloidiasis in Indigenous health in Australia.

## Supporting information

**S1 Table. *Strongyloides* reports extracted at half-yearly intervals over 4.5 years in four clinics in remote Arnhem Land (July 2012 to December 2016).**
(DOCX)

**S2 Table. Analysis for each clinic and overall, for 680 persons with at least one positive test.** 2012–2016.
(DOCX)

**S3 Table. Breakdown for each clinic for reinfection and number positive on last test.** 2012–2016.
(DOCX)

**S4 Table. Final Evaluation 2012–2020.**
(DOCX)

## Acknowledgments

We thank the board of Miwatj Health Aboriginal Corporation, the staff and community members for their encouragement and support. We also thank Petra Buettner, Steve White, and Jenny Shield for their expert contributions. We honour our teacher the late Emeritus Professor Rick Speare who inspired us with the vision for a national strongyloidiasis control program to eliminate strongyloidiasis in remote Indigenous communities in Australia.

## Author contributions

**Conceptualization:** Wendy A Page.

**Data curation:** Wendy A Page.

**Formal analysis:** Wendy A Page, Karen Dempsey.

**Investigation:** Wendy A Page.

**Methodology:** Wendy A Page.

**Project administration:** Wendy A Page.

**Resources:** Wendy A Page.

**Supervision:** David Blair, Jenni A Judd.

**Validation:** Karen Dempsey, Jenni A Judd.

**Visualization:** Wendy A Page, David Blair, Karen Dempsey.

**Writing – original draft:** Wendy A Page, David Blair, Jenni A Judd.

**Writing – review & editing:** Wendy A Page, David Blair, Karen Dempsey, Beverley-Ann Biggs, Jenni A Judd.

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
