## [Decision Letter · Decision Letter 0]

PNTD-D-24-01494Evaluating the effectiveness and sustainability of a primary healthcare strategy to reduce the prevalence of strongyloidiasis in endemically infected Indigenous communities in northern Australia.PLOS Neglected Tropical Diseases Dear Dr. Page, Thank you for submitting your manuscript to PLOS Neglected Tropical Diseases. After careful consideration, we feel that it has merit but does not fully meet PLOS Neglected Tropical Diseases's publication criteria as it currently stands. Therefore, we invite you to submit a revised version of the manuscript that addresses the points raised during the review process. Please submit your revised manuscript within 30 days Apr 15 2025 11:59PM. If you will need more time than this to complete your revisions, please reply to this message or contact the journal office at plosntds@plos.org. Please include the following items when submitting your revised manuscript: * A rebuttal letter that responds to each point raised by the editor and reviewer(s). You should upload this letter as a separate file labeled 'Response to Reviewers'. This file does not need to include responses to any formatting updates and technical items listed in the 'Journal Requirements' section below. * A marked-up copy of your manuscript that highlights changes made to the original version. You should upload this as a separate file labeled 'Revised Manuscript with Track Changes'. * An unmarked version of your revised paper without tracked changes. You should upload this as a separate file labeled 'Manuscript'. If you would like to make changes to your financial disclosure, competing interests statement, or data availability statement, please make these updates within the submission form at the time of resubmission. Guidelines for resubmitting your figure files are available below the reviewer comments at the end of this letter. We look forward to receiving your revised manuscript. Kind regards, Eduardo José Lopes-Torres, Ph.D.Academic EditorPLOS Neglected Tropical Diseases Krystyna CwiklinskiSection EditorPLOS Neglected Tropical Diseases

Shaden Kamhawi

co-Editor-in-Chief

Paul Brindley

co-Editor-in-Chief

**Reviewers' comments:** Reviewer's Responses to Questions

**Key Review Criteria Required for Acceptance?**

**Methods**:

-Are the objectives of the study clearly articulated with a clear testable hypothesis stated?

-Is the study design appropriate to address the stated objectives?

-Is the population clearly described and appropriate for the hypothesis being tested?

-Is the sample size sufficient to ensure adequate power to address the hypothesis being tested?

-Were correct statistical analysis used to support conclusions?

-Are there concerns about ethical or regulatory requirements being met?

Reviewer #1: All questions and criteria are met in the manuscript.

Reviewer #2: It would be good to provide more information about the ELISA testing. How were results standardized over time? What was the turnaround time before results were returned to the clinic?

**Results**:

-Does the analysis presented match the analysis plan?

-Are the results clearly and completely presented?

-Are the figures (Tables, Images) of sufficient quality for clarity?

Reviewer #1: All questions and criteria are met in the manuscript.

Reviewer #2: Are the data sufficient to look at incidence?

What do the communities think about the project?

**Conclusions**:

-Are the conclusions supported by the data presented?

-Are the limitations of analysis clearly described?

-Do the authors discuss how these data can be helpful to advance our understanding of the topic under study?

-Is public health relevance addressed?

Reviewer #1: All questions and criteria are met in the manuscript.

Reviewer #2: What has been done to scale up the intervention to other communities?

Why doesn’t Australia procure WHO prequalified generic ivermectin?

**Editorial and Data Presentation Modifications?**

Reviewer #1: None.

Reviewer #2: The manuscript is well written.

**Summary and General Comments**:

Reviewer #1: The manuscript "Evaluating the effectiveness and sustainability of a primary healthcare strategy to reduce the prevalence of strongyloidiasis in endemically infected Indigenous communities in northern Australia" describes the use of serology for Strongyloides combined with treatment as a strategy for controlling strongyloidiasis in an endemic area with an Indigenous population. The United Nations 2030 Agenda for Sustainable Development aims to end epidemics of AIDS, tuberculosis, malaria, and tropical diseases, such as soil-transmitted helminthiasis. However, within the context of soil-transmitted helminthiasis, efforts are primarily focused on ascariasis, trichuriasis, and hookworm, with little attention given to the control of strongyloidiasis. In the manuscript from northern Australia, the authors present an experience report that shows a reduction in disease rates, which seems to be a promising strategy, though it requires further follow-up to be validated.

The manuscript is written in a simple and straightforward manner, and although the data presented in this paper are linked to a previous publication, it is still understandable.

Below are some considerations regarding the manuscript:

1. The data in Table 1 and S1 Table need to be reviewed. This also impacts the specific values described in the sections "Final evaluation 2012–2020" and "Analysis of data from the implementation phase 2012–2016."

2. Were the 13 individuals who had a negative diagnosis in the previous serology but tested positive in a new test, treated? Was there therapeutic success?

3. Were the 17 individuals who remained serologically positive after treatment retreated? What was the protocol for cases where individuals remained positive throughout the periodic testing?

4. In endemic areas, the main hypothesis for persistent infection is reinfection, but we cannot exclude the possibility of parasitic resistance, especially considering the limited therapeutic options available for the treatment of these infections. Therefore, it is important to discuss this issue in the manuscript, and to acknowledge the inability to differentiate between reinfection and resistance as a limitation of the study.

5. The authors state: "Persons who stayed positive were significantly younger (median age 31 years; IQR = (21, 42)) compared with persons who became negative (median age 39 years; IQR = (28, 51); p<0.001; Mann-Whitney test). Reinfected persons were on average older (median age 42 years; IQR = (31, 50)) compared to persons with no reinfection (median age 35 years; IQR = (24, 47); p=0.032; Mann-Whitney test)." Is there any hypothesis to explain this finding? This data was not well discussed.

6. Were health education strategies employed during the study? Typically, education activities are crucial for reducing prevalence in endemic areas where selective treatment is carried out.

Reviewer #2: In this manuscript, Page and colleagues report on the sustainability of a Strongyloides control program implemented through the routine health care system in indigenous communities in Australia. The results show that this strategy results in a dramatic reduction in disease prevalence over time.

PLOS authors have the option to publish the peer review history of their article (what does this mean?). If published, this will include your full peer review and any attached files.

Reviewer #1: **Yes: **Maria Fantinatti

Reviewer #2: No

---

## [Decision Letter · Decision Letter 1]

Dear Dr Page,

We are pleased to inform you that your manuscript 'Evaluating the effectiveness and sustainability of a primary healthcare strategy to reduce the prevalence of strongyloidiasis in endemically infected Indigenous communities in northern Australia.' has been provisionally accepted for publication in PLOS Neglected Tropical Diseases.

Best regards,

Eduardo José Lopes-Torres, Ph.D.

Academic Editor

Krystyna Cwiklinski

Section Editor

Shaden Kamhawi

co-Editor-in-Chief

Paul Brindley

co-Editor-in-Chief

Reviewer's Responses to Questions

**Key Review Criteria Required for Acceptance?**

**Methods**

-Are the objectives of the study clearly articulated with a clear testable hypothesis stated?

-Is the study design appropriate to address the stated objectives?

-Is the population clearly described and appropriate for the hypothesis being tested?

-Is the sample size sufficient to ensure adequate power to address the hypothesis being tested?

-Were correct statistical analysis used to support conclusions?

-Are there concerns about ethical or regulatory requirements being met?

Reviewer #1: (No Response)

Reviewer #2: (No Response)

**Results**

-Does the analysis presented match the analysis plan?

-Are the results clearly and completely presented?

-Are the figures (Tables, Images) of sufficient quality for clarity?

Reviewer #1: (No Response)

Reviewer #2: (No Response)

**Conclusions**

-Are the conclusions supported by the data presented?

-Are the limitations of analysis clearly described?

-Do the authors discuss how these data can be helpful to advance our understanding of the topic under study?

-Is public health relevance addressed?

Reviewer #1: (No Response)

Reviewer #2: (No Response)

**Editorial and Data Presentation Modifications?**

Reviewer #1: (No Response)

Reviewer #2: (No Response)

**Summary and General Comments**

Reviewer #1: (No Response)

Reviewer #2: (No Response)

PLOS authors have the option to publish the peer review history of their article (what does this mean?). If published, this will include your full peer review and any attached files.

Reviewer #1: **Yes: **Maria Fantinatti

Reviewer #2: No

---

## [Editor Report · Acceptance letter]

Dear Dr Page,

We are delighted to inform you that your manuscript, "Evaluating the effectiveness and sustainability of a primary healthcare strategy to reduce the prevalence of strongyloidiasis in endemically infected Indigenous communities in northern Australia.," has been formally accepted for publication in PLOS Neglected Tropical Diseases.

Best regards,

Shaden Kamhawi

co-Editor-in-Chief

Paul Brindley

co-Editor-in-Chief
